# Incidence, Etiology, Prevention and Management of Ureteroenteric Strictures after Robot-Assisted Radical Cystectomy: A Review of Published Evidence and Personal Experience

**Shintaro Narita ***, **Mitsuru Saito, Kazuyuki Numakura and Tomonori Habuchi**

Department of Urology, School of Medicine, Akita University, Akita 010-8543, Japan; urosaito@gmail.com (M.S.); numakura@doc.med.akita-u.ac.jp (K.N.); thabuchi@gmail.com (T.H.)

**\*** Correspondence: naritashintaro@gmail.com; Tel.: +81-18-884-6154; Fax: +81-18-836-2619

**Abstract:** Benign ureteroenteric anastomosis strictures (UESs) are one of many critical complications that may cause irreversible disability following robot-assisted radical cystectomy (RARC). Previous studies have shown that the incidence rates of UES after RARC can reach 25.3%, with RARC having higher UES incidence rates compared to open radical cystectomy. Various known and unknown factors are involved in the occurrence of UES. To minimize the incidence of UES after RARC, our group has standardized the procedure and technique for intracorporeal urinary diversion by applying the following five strategies: (1) wide delicate dissection of the ureter and preservation of the periureteral tissues; (2) gentle handling of the ureter and security of periureteral tissues at the anastomotic site; (3) use of indocyanine green to confirm good blood supply; (4) standardization of the ample ureteral spatulation length for Wallace ureteroenteric anastomosis through objective measurements; and (5) development of an institutional standardized procedure manual. This review focused on the incidence, etiology, prevention, and management of UES after RARC to bring attention to the incidence of this complication while also proposing standardized surgical procedures to minimize its incidence after RARC.

**Keywords:** robot assisted radical cystectomy; urothelial cancer; ureteroenteric stricture

## 1. Introduction

Despite becoming a standard surgical procedure for the definitive treatment of patients with localized muscle invasive bladder cancer, robot-assisted radical cystectomy (RARC) remains complex with a non-negligible learning curve and numerous complications [1]. The RAZOR trial, which was conducted to compare RARC with open radical cystectomy (ORC), showed no difference in three-year progression-free survival between both procedures, with RARC exhibiting significant advantages in estimated blood loss, blood transfusion rates, and length of stay [2,3]. In terms of the incidence of complications, several randomized trials have demonstrated that both RARC and ORC have comparable incidence rates of common complications [2,4,5]. However, reported complication rates of RARC have ranged from 30% to 70%, suggesting the urgent needed for intra- and peri-operative strategies to reduce complications [6]. RARC has been known to have lower rates of severe short-term complications within 90 days after surgery [7]. However, its long-term complications have been relatively high-grade, with insufficient evidence available to make definitive conclusions on the long-term complications of RARC [8,9]. One of the short- and long-term complications is a benign ureteroenteric anastomosis stricture (UES), a critical complication that can cause irreversible disability such as chronic kidney dysfunction. This review focuses on the incidence, etiology, prevention, and management of UES after RARC and aims to propose standardized surgical steps to reduce the incidence of this complication.

While the initial management of UES through endoscopic and percutaneous techniques may be feasible, high success rates have been observed only with surgical repair.

## 2. Incidence

A benign UES is a known complication of radical cystectomy (RC) and urinary diversion with an incidence rate ranging from 3% to 10% [10–14]. Accumulating evidence from studies focusing on UES has suggested that 6.5%–25.3% of patients who undergo RARC develop UES [15–21] (Table 1). However, the duration of follow-up and definition of UES in the previous literature vary. Although most of the studies have utilized obstruction on radiographic imaging to define UES, the presence of an "obstruction" is generally determined based on the physician's expertise. Moreover, studies reporting on overall complications tend to have lower incidence rates of UES compared to those focusing on UES [21]. Follow-up duration and timing of evaluation have also been associated with the reported incidence of UES given a very relevant clinical fact: 75% of patients with UES are asymptomatic [16,17,22].

Among five available studies comparing UES rates between RARC and ORC [15,16,18–21,23], four showed that RARC promoted higher UES incidence rates compared to ORC, whereas one showed that RARC had lower rates of UES [20]. Moreover, two studies concluded that RARC had a significantly higher UES rates compared to ORC [18,21]. An initial single-center series including patients who received RC between 2007 and 2011 showed that RARC with ECUD had higher UES rates (12.6%) compared to ORC (8.5%) [15]. Moreover, we should be aware of the fact that urinary diversion in the vast majority of patients reviewed in previous comparative studies was performed in a mini-open incisional fashion (ECUD). Reesink et al., who conducted a retrospective study to assess the incidence of UES after ORC and RARC with ICUD performed at a single center between 2012 and 2017 [21], found a total UES incidence rate of 16.8% during the median follow-up of 50 months. The authors demonstrated that RARC (25.3%) had a significantly higher incidence rate of UES compared to ORC (13.0%). Moreover, the UES rate was high (47%) during the first year after RARC and among those in the left-sided group (78.7%). However, the rate of UES in the study was relatively high even in the ORC group, suggesting that specific operative procedures in the institute might have influenced the outcomes. By contrast, Hosseini et al. claimed that the overall incidence of UES was low (6.5%) in a retrospective observational study on 371 patients who underwent RALC with ICUD, although the study also found that the left-side stricture (63%) was higher than the right-side (29%) and bilateral stricture (8%) [17]. Considering that the left ureter requires more extensive dissection and mobilization and tunneling under the sigmoid mesentery, higher incidences of UES are observed in the left ureter [16,24]. Taken together, particular care should therefore be taken for UES prevention and diagnosis when dealing with the left ureter and in the induction period of RARC.

**Table 1.** Studies focusing on the incidence of ureteroenteric stricture after RARC.

| Author | Year | Number of Patients | ICUD or ECUD | Urinary Diversion in RARC | Anastomosis | Definition of UES | Time to UES (Mean) | Results |
|---|---|---|---|---|---|---|---|---|
| Anderson (2013) | 2007–2011 | 103 RARC vs. 375 ORC | ECUD | IC 91.3%, NB 8.7% | Bricker | Imaging and intervention | 5.3 m | 12.6% in RARC vs. 8.5% in ORC, $P = 0.21$ |
| Ahmed (2017) | 2005–2016 | 440 RARC | ECUD or ICUD | IC | NA | Obstruction on imaging | 5 m | 12%,16%,19% at 1, 3, 5yrs |
| Hosseini (2018) | 2003–2015 | 371 RARC | ICUD | IC 65%, NB 35% | Wallace | Obstruction on imaging | 165 days | 6.5% |
| Goh (2019) | 2009–2014 | 332 RARC vs. 1449 ORC | NA | Incontinent 84%, continent 5.1%, unknown 10.8% | NA | Need for Intervention | NA | at 6 m, 1yr, 2yrs: 9%, 11.6%, 13.9% in RARC vs. 4.2%, 7.4%, 8.3% in ORC, $P < 0.05$ |
| Ericson (2020) | 2011–2018 | 689 RARC vs. 279 ORC | ECUD or ICUD | IC 77.5%, NB 12.2% | Bricker | Obstruction on imaging | 4.7–5.1 m | 11.3 in ECUD-RARC vs. 13.0% in ICUD-RARC vs. 9.3% in ORC, $P = 0.37$ |
| Faraj (2021) | 2007–2018 | 39 RARC-ICUD vs. 297 RARC-ECUD vs. 337 ORC | ECUD or ICUD | NB 14.8% | Bricker or Wallace | Obstruction on imaging | 5 m | 2.6% in RARC-ICUD, 9.6% in RARC-ECUD, 8.0% in ORC, $P = 0.33$ |
| Reesink (2021) | 2012–2017 | 87 RARC vs. 192 ORC | ICUD | IC 91.8% or NB 8.2% (including open) | Bricker | Obstruction on imaging | 3.0 m | 25.3% in RARC vs. 13.0% in ORC, $P = 0.015$ |

Abbreviation: ICUD;intracorporeal urinary diversion, ECOD;extracorporeal urinary diversion, UES; ureteroenteric stricuture, IC; ileal conduit, NB; orthotopic neobladder, RARC;robot assisted radical cyctectomy, ORC; open radical cyctectomy, NA; not assessed.

Regarding the anastomotic technique, a recent single-center series on ORC showed that the Bricker technique (25.3%) promoted significantly higher rates of UES compared to the Wallace technique (7.7%) [25]. However, a previous meta-analysis including RARC showed no differences in the rate of UES between the Wallace and Nesbit/Bricker techniques [26]. Furthermore, which type of urinary diversion, including ECUD or ICUD, is superior still remains controversial. Excessive ureteral handling may be associated with higher incidences of UES in both ECUD and ICUD, but ICUD may be associated with less traumatic maneuvers (e.g., excessive stretching) of the ureters and a more accurate surgical technique thanks to magnification. A study by Faraj et al., which was conducted to investigate UES rates with RARC-ECUD or RARC-ICUD, noted that ICUD had a lower UES incidence (2.6%) compared to ECUD (6.4%) [20]. However, the impact of ICUD or ECUD on the incidence rates of UES remains unclear considering the lack of large-scale randomized prospective studies.

## 3. Etiology

Most of the retrospective studies suggest that multiple factors are associated with UES and its prevention after RARC. First of all, the surgeon's experience seems to be strongly associated with the incidence of UES [21], given that several studies showed higher rates of UES in initial introductory cases at each institution [16,21]. Given the nature of robotic surgery, less haptic feedback and magnified visualization can cause excessive handling of the ureters. Compromised vascularity of the ureters can also be potentially associated with UES [16]. Ahmed et al. emphasized the importance of sound surgical technique, including adequate ureteral dissection while maintaining sufficient adventitia, avoiding cauterization, wide spatulation, and a watertight anastomosis that is not under tension [16], which are rather difficult to evaluate objectively [16]. A study by Yuh et al. on 14 patients with UES among a total of 241 consecutive patients who underwent RARC revealed that inadvertent kinking or twisting of the ureters and/or diversion might occur, causing urinary diversion-related complications [27]. The type of anastomosis (running or interrupted sutures), the length of ureters, and the adoption of an antireflux technique may also influence the incidence of UES. A previous study demonstrated that patients with postoperative anastomotic urinary leakage had approximately four times higher rates of UES compared to those who did not [16]. Given that urinary tract infections (UTIs) impair healing and cause scarring with the release of inflammatory mediators and proteases, UTIs may be associated with the presence of UES. Moreover, preoperative kidney function and nutritional status may be considered potential risk factors for UES [16].

A population-based study using a surveillance, epidemiology, and end results program demonstrated that RARC (vs. ORC) and preoperative hydronephrosis were significantly associated with the development of UES [18]. Apart from the aforementioned factors, multivariable analysis in a single-center study involving 440 RARC cases with a 13% incidence of UES showed that body mass index, intracorporeal urinary diversion, length of the right resected ureter, estimated glomerular filtration rate 30 days after RARC, urinary tract infection, and leakage were independent predictors for UES [16]. Taken together, the aforementioned studies strongly suggest that various known and unknown factors are involved in the occurrence of UES.

## 4. Prevention

To date, no confirmatory standardized surgical techniques or procedures have been established to completely prevent UES. Ahmadi et al. proposed a novel technique to decrease its incidence after RARC using indocyanine green (ICG) with near-infrared fluorescence during surgery [28]. Notably, they showed that the incidence of UES decreased from 10.6% to 0% after utilizing the technique. Similarly, another study found that the ICG group had a UES incidence rate of 0%, whereas the non-ICG group had an incidence rate of 7.5% [29]. The study also mentioned that 34% of the ureters had poor distal blood perfusion requiring more proximal dissection [29]. Based on these findings, our institution had also

adopted this technique starting February 2020. Briefly, ICG is prepared by mixing 25 mg of dye in 10 mL of distilled water. Just before ureteroenteric anastomosis, 5 mL of ICG was intravenously injected, which allowed the near-infrared fluorescence system of the da Vinci Xi surgical system® to observe its fluorescence (Figure 1B). Thereafter, a portion of the non-fluorescent ureter was dissected and discarded before anastomosis, given the strong possibility of poor blood supply in such a portion (Figure 1B). On the other hand, Ahmed et al. proposed several techniques, such as a longer ileal conduit to minimize tension to the ureteroenteric anastomosis, creating a buttonhole-like instead of a slit-like enterotomy, which results in a wider area of anastomosis and retroperitonization to promote healing and prevent UES [16]. Given the strong association between a surgeons' experience and UES incidence, structured training programs with information regarding standardized techniques is imperative, particularly for novice surgical teams.

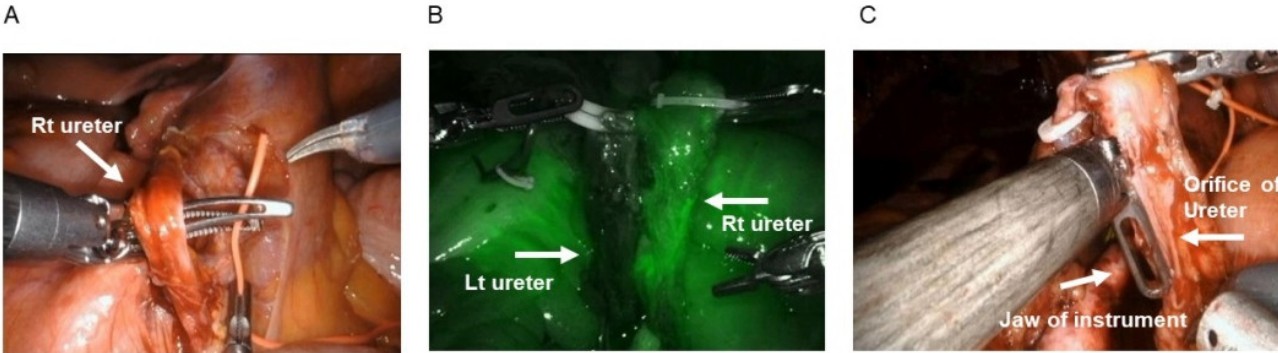

**Figure 1.** Standardized techniques to avoid UES. (**A**) preservation of the periureteral tissues in the ureter, (**B**) use of ICG, and (**C**) standardization of the ureteral spatulation length (around 1 inch) for ureteroenteric anastomosis via objective measurements with a jaw length of da Vinci instruments.

Detailed surgical techniques of ICUD were summarized in the previous literatures [30,31]. We have adopted ICUD with the Wallace anastomosis for both ileal conduit and ileal neobladder because of the wider anastomotic caliber in this method that than in the Bricker method based on the relatively high rates of UES in our initial series, we have standardized the procedure and technique for ICUD urinary diversion to minimize the incidence of UES after RARC by applying the following five strategies: (1) wide delicate dissection and preservation of periureteral tissues to secure good blood supply (Figure 1A); (2) gentle handling of the ureter and security of periureteral tissues at the anastomotic site; (3) use of ICG to visualize ureteral vascularity and confirm good blood supply (Figure 1B); (4) standardization of the ureteral spatulation length (around 1 inch) for Wallace ureteroenteric anastomosis through objective measurements (Figure 1C); and (5) development of the institutional standardized procedure manual. Briefly, at the very beginning of all RARC procedures, the ureters were secured with adequate periureteral tissue. The length of forceps may be helpful to objectively measure the length of ureteral spatulation (Figure 1B), allowing us to standardize the length of ureteral orifice. We adopted a spatulated length of at least 1 inch for the ureteral orifice in the light of previous literature and diameter of the ileum [17,32]. After retrospectively reviewing the medical records of patients with MIBC treated with RARC between 2013 and 2021, we compared the incidence of UES before (January 2020) and after (February 2020) the adoption of the aforementioned standardization techniques (Table 2). A total of 24 and 17 patients with ICUD urinary diversion were included in the non-standardized and standardized groups, respectively. The rates of ileal neobladder in the non-standardized and standardized groups of RARC with ICUD were 58.3% (14 pts) and 41.2% (7 pts), respectively. Notably, the standardized group had significantly lower incidence rates of grade $\leq$ 2 hydronephrosis (9.1%; 3/33 ureters vs. 30.4%; 14/46 ureters, $P = 0.023$) and grade $\geq$ 3 ureteral complications (0.0%; 0/33 ureters vs. 13.0%; 6/46 ureters, $P = 0.031$) compared to the non-standardized group 1 month after

surgery during the follow-up. Our results suggested that the standardization of ICUD procedures in RARC has the potential to reduce the risk of UES, although larger studies with longer follow-up are needed to confirm our findings.

**Table 2.** The rates of UES before and after standardization of surgical procedure in our institution.

| Surgical procedure | Before Standardization | | | After Standardization | | | *P* value (≤G2) | *P* value (G3) |
|---|---|---|---|---|---|---|---|---|
| | Number of ureters | ≤G2 at 1 m | G3 during follow-up | Number of ureters | ≤G2 at 1 m | G3 during follow-up | | |
| ALL RARC-IC or NB | 32 | 15 (46.9%) | 4 (12.5%) | 17 | 2 (11.8%) | 0 (0%) | 0.014 | 0.128 |
| ICUD-IC or NB | 24 | 11 (45.8%) | 4 (16.7%) | 17 | 2 (11.8%) | 0 (0%) | 0.021 | 0.076 |

G2: grade 2, G3: grade 3, RARC: robot-assisted radical cyctectomy, IC: ileal conduit, NB: neo-bladder, ICUD: intracorporeal urinary diversion.

## 5. Management

Although no standardized therapies have been established for the treatment of UES after RARC, open revision had been the gold standard management of UES after urinary diversion due to its higher success rate as compared to the endoscopic approach [16]. However, open revision is generally challenging and has been accompanied by a high risk of additional complications [33,34]. Therefore, initial management of UES via endoscopic or percutaneous techniques may be attempted. One study including 58 patients with UES after RC showed that endoscopic intervention succeeded in 51.3% of the patients [35]. On the other hand, 78% of the 32 patients who underwent open revision via direct implantation or tissue interposition (six Boari flaps and seven ileal segments) achieved long-term success [35]. Another retrospective study on 41 patients with UES after RC found an 87% success rate for open revision [36]. The same study also stated that the addition of the chimney modification to the orthotopic neobladder facilitated surgical repair [36]. In cases with very severe bilateral strictures in ICUD-neobladder, Rayn et al. proposed a technique called "Reverse 7," wherein the ileal segment is anastomosed to the bilateral renal pelvis on each side and then directly anastomosed to the top of the neobladder [37]. Robotic repair has also been considered as an option for the management of UES. However, evidence is scarce on this topic [38,39].

Ahmed et al. summarized the treatment of UES after RARC [16]. Accordingly, all 51 patients were initially treated with endoscopic and percutaneous approach, including 29 (57%) who underwent endoscopic and percutaneous management alone and 22 (43%) who required additional open (6 patients) or robotic (16 patients) surgical treatment. After a median follow-up of 23 months, 33 patients (65%) were free of disease, among whom 13 received endoscopic or percutaneous repairs, 15 received robot-assisted repairs, and 5 received open revisions. The authors also noted that open and robot-assisted revisions had a 100% success rate with the intraoperative complication (serosal tears) in two patients in the robot-assisted group [16]. With regard to risk factors for failure of UES treatment, male gender and higher BMI were reported to be associated with lower odds of successful endoscopic management.

Although the advantage of robotic repair over open repair remains unclear, robotic repair is an attractive option for the treatment of UES. Gin et al. reported the outcomes of 41 patients who underwent UES repair between 2007 and 2015, among whom 11.9% received the robotic approach [40]. The study showed that a 100% success rate was achieved without any re-operation during the median follow up of 16.3 months in a total of 50 renal units [40]. Tobis et al. reported that all four patients with UES after RC [38] were successfully repaired via the robotic approach, with no complications after a mean follow-up duration of 16 months [38]. A retrospective study comparing robotic repair (*n* = 7) and open repair (*n* = 5) in patients with UES after RC, including five RARC, showed that both approaches had comparable median estimated blood loss, operative time, and hospital stay [41]. Furthermore, three patients developed complications in the open group,

whereas no complications were observed in the robot group [41]. During robotic repair, Tuderti et al. highlighted the usefulness of ICG injection via nephrostomy to identify the healthy ureter [42]. Kaouk et al. successfully introduced da Vinci SP®, which is a single port platform, to treat three patients with UES, including one with bilateral UES after cystectomy [43]. Further validation studies on the usefulness of a conventional robot-assisted system or a single port robotic system are therefore required for repair of UES, especially for challenging cases with UES.

### 6. Conclusions

This review highlights the profound need to pay special attention to the prevention, early diagnosis, and meticulous repair of UES along with the appropriate follow-up of patients with UES after RARC. Careful establishment and implementation of standardized procedures, techniques, tips and tricks, and retrospective review of personal data and experience are crucial in detecting key reproducible points to reduce the incidence of UES, which is a devastating clinical complication after RARC.

**Author Contributions:** Conceptualization, S.N.; methodology, S.N.; resources, S.N., M.S., K.N. and T.H.; writing—original draft preparation, S.N.; supervision, T.H. All authors have read and agreed to the published version of the manuscript.

**Funding:** This research received no external funding.

**Institutional Review Board Statement:** The study was conducted according to the guidelines of the Declaration of Helsinki, and approved by the Ethics Committee of Akita University School of Medicine (protocol code 1517, and 11 April 2016).

**Data Availability Statement:** The data presented in this study are available on request from the corresponding author. The data are not publicly available due to privacy and ethical reasons.

**Conflicts of Interest:** The authors declare no conflict of interest.

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
