# Peer review of "Incidence, Etiology, Prevention and Management of Ureteroenteric Strictures after Robot-Assisted Radical Cystectomy: A Review of Published Evidence and Personal Experience"

_curroncol, doi:10.3390/curroncol28050348_

Round 1

Reviewer 1 Report

Robotic cystectomies are increasing worldwide,nwith complication rates of 30-70% reported. UES are major complications, a reduction is certainly desirable here, which is why a review will be very helpful.

The introduction correctly refers to the results of the RAZOR trial. As in the other randomized studies (e.g. Bochner et al.), no significant differences in terms of complications between RARC and ORC were found. It should be noted, however, and must be taken into account in the discussion that in all randomized studies published to date, urinary diversion after robotic cystectomy was carried out openly (ECUD). Thus, no difference was to be expected in the complications that are mainly associated with the urinary diversion. The urinary diversion is open in both arms, with the ECUD certainly via a smaller incision, so that if a difference were found, this group had to do worse.

The study with 25.3% UES was carried out as an ICUD, but here too the number of UES in the ORC group is very high at 13.0%. It can be assumed that there is a methodological problem in the group, which is why standardization, as described in the present work, is necessary. This must be addressed.

The most important factor in this type of complication is the surgeon and his or her experience. I think if a statement should be made about the method, the operator's experience must come first. This should be discussed on the basis of the literature (experience and how many surgeons).

When recommending the procedure (5 points), points 1 and 2 are understandable and general. Point 3 (ICG) is explained in the paper through literature. Point 4 is not discussed and explained. How do you get this length? Are there any information / results on this in the literature?

In summary, the review is helpful for everyone who conducts ICUD. However, the individual points that should be standardized must all be explained and discussed in the paper. In addition, it should come out that most of the papers compare ECUD with ORC. One of the largest works with ICUD (Hosseini 2018) only demonstrated UES at 6.5% and is therefore in the range of the ORC.

Author Response

Oct 7, 2021

Prof. Sharlene Gill, MD, PhD,

Editor-in-Chief

Current Oncology,

Manuscript ID: curroncol-1389145
Type of manuscript: Review
Title: Incidence, etiology, prevention, and management of ureteroenteric
strictures during robot-assisted radical cystectomy: A review
Authors: Shintaro Narita *, Mitsuru Saito, Kazuyuki Numakura, Tomonori Habuchi

Dear Prof. Sharlene Gill,

Thank you very much for your email of 22-Sep-2021, with regard to our manuscript (curroncol-1389145) together with your reviewers' comments.

The comments of the reviewers have been helpful in allowing us to revise our manuscript. We have revised the manuscript point by point as follows and the manuscript has been shaped up by an English editing service again.

Reviewer 1

Robotic cystectomies are increasing worldwide, with complication rates of 30-70% reported. UES are major complications, a reduction is certainly desirable here, which is why a review will be very helpful.

The introduction correctly refers to the results of the RAZOR trial. As in the other randomized studies (e.g. Bochner et al.), no significant differences in terms of complications between RARC and ORC were found. It should be noted, however, and must be taken into account in the discussion that in all randomized studies published to date, urinary diversion after robotic cystectomy was carried out openly (ECUD). Thus, no difference was to be expected in the complications that are mainly associated with the urinary diversion. The urinary diversion is open in both arms, with the ECUD certainly via a smaller incision, so that if a difference were found, this group had to do worse.

Answer: I would like to thank the reviewer for the important comments. As the reviewer pointed out, previous comparative studies included a considerable number of patients with ECUD. Therefore, we have added a comment in the Incidence section (Page 4, Line 66–68).

The study with 25.3% UES was carried out as an ICUD, but here too the number of UES in the ORC group is very high at 13.0%. It can be assumed that there is a methodological problem in the group, which is why standardization, as described in the present work, is necessary. This must be addressed.

Answer: We agree with the reviewer’s opinion that the rate of UES was too high in the ORC group of the study reported by Reesik et al. In addition, the rate of UES was low in another ICUD series as discussed below. Therefore, we have added some additional comments regarding the issue in the Incidence section (Page 4, Line74–79).

The most important factor in this type of complication is the surgeon and his or her experience. I think if a statement should be made about the method, the operator's experience must come first. This should be discussed on the basis of the literature (experience and how many surgeons).

Answer: We thank the reviewer’s helpful comment. As the reviewer recommended, we have added the sentence regarding the impact of surgeon’s experience on the incidence of UES and some discussions about it in the first sentence of the Etiology section (Page 4, Line 99–101).

When recommending the procedure (5 points), points 1 and 2 are understandable and general. Point 3 (ICG) is explained in the paper through literature. Point 4 is not discussed and explained. How do you get this length? Are there any information / results on this in the literature?

Answer: We completely understand the reviewer’s concern about supporting literatures for standardization of the ureteral spatulation length. Previous studies have proposed 2cm spatulation of the ureter for the Wallace anastomosis (Hosseini et al,2011, PMID: 21917098, Kavaric et al, 2020, PMID: 32167712); however, we believe that the appropriate spatulation was wider than 2cm when considering the diameter of ileum. On the basis of it, we decided to propose the “1 inch” spatulation length of the ureter. We have modified the sentence in the Prevention section (Page 6, Line 170–171)

In summary, the review is helpful for everyone who conducts ICUD. However, the individual points that should be standardized must all be explained and discussed in the paper. In addition, it should come out that most of the papers compare ECUD with ORC. One of the largest works with ICUD (Hosseini 2018) only demonstrated UES at 6.5% and is therefore in the range of the ORC.

Answer: We appreciate that the reviewer gave us the important comments. As mentioned above, we have added some discussions regarding the results of the largest series with ICUD (Page 4, Line 74–79).

Reviewer 2

This paper is interesting and brings attention to a very challenging and complex condition, such as UES after robotic radical cystectomy.  I like the fact that the title reports the 4 topics around which the paper is organised (Incidence, Etiology, Prevention and Management of UES), but I would change the wording: it is not DURING RARC, but AFTER RARC. Moreover the paper, presented as a review in the title, is not only a review, it reports also the original experience of the authors. So, as a first thing, I would change the title, and make it more correspondent to the text: INCIDENCE, ETIOLOGY, PREVENTION AND MANAGEMENT OF UES AFTER RARC: REVIEW OF PUBLISHED EVIDENCE AND PERSONAL EXPERIENCE.

Answer: We would like to thank the reviewer for the helpful suggestion. As the reviewer suggested, we have modified the title of the manuscript (Page 1, Line 3–4).

A number of changes to the text are needed before the paper can be considered for publication, as shown below:

Answer: Thank the reviewer for blushing up the manuscript. We have corrected some sentences according to the reviewer’s recommendation and suggestion from an English editing service.

Abstract, line 9: Robot-assisted radical cystectomy (not prostatectomy)

                 line 10: ...can reach 25.3% of cases

Page 1, line 38: definitive conclusions

              line 41: this review focuses...

              line 42: ...after RARC, with the final aim to draw attention to this                                      severe clinical complication and to propose standardised                                    surgical steps able to reduce its incidence.

              line 44: may be possible/feasible (not ideal)

              line 45: ...have been observed only with surgical repair

Page 2, line 48: ...with an incidence rate ranging from 3 to 10%.

              line 56: ...given a very relevant clinical fact: 75% of patients with                                  UES are asymptomatic

Page 4, line 75: Particular care should therefore be taken with UES                                               prevention and diagnosis when dealing with the left ureter                                 and in the first months after RARC.

Page 4, line 84: but ICUD may be associated with less stretching of the                                        ureters and more accurate surgical technique thanks to                                      magnification

Page 4, line 86: ...the incidence rates of UES remain unclear considering the                             lack of...

Answer: We appreciate the reviewer’s appropriate guidance for blushing up our manuscript. We have modified the manuscript according to the suggestion.

Page 5, line 33 and following: why there is no mention on proposed standardised techniques in case of orthopic neobladders? Do the authors have experience on this? How should the anastomosis be done? How long the spatulation of the ureters in this case? Which technique? Nothing to add on this point? Or is the paper mainly focused on ileal conduits? If so, this should be specified in the title. Otherwise, the authors should mention trips and tricks in case of orthotopic neobladders, published evidence and personal experience.

Answer: We appreciate that the reviewer recommends us to add our standardized technique for orthotopic neobladder and surgical techniques in our institute. We adopt ICUD with the Wallace anastomosis for both ileal conduit and ileal neobladder. Given the manuscript focuses on the technique of ureteroenteric anastomosis during ICUD, we have introduced previous literatures focusing on standard surgical techniques of ICUD and described the current situation for surgical techniques in our institution (Page 6, Line 155–157).

Page 5, line 47: RARC (not RALC)

              line 49: and preservation of periureteral tissues...

Answer: We have corrected the sentences.

Page 6, line 52: a Total of 24 and 12 patients with ICUD urinary diversion                                     (PLEASE SPECIFY WHICH ONE: ILEAL CONDUIT?                                                NEOBLADDER?). THIS IS A VERY CRUCIAL POINT.

Answer: Thank the reviewer for asking us the number of patients treated with ileal conduit or ileal neobladder in our series. The rates of ileal neobladder in the non-standardized and standardized groups of RARC with ICUD were 58.3% (14 pts) and 41.2% (7pts), respectively. We have added the result in the Prevention section (Page 6, Line 175–178).

              line 73:...due to its higher success rate as compared...

Answer: Thank the reviewer for the helpful suggestion. We have modified the sentence.

              line 76: YOU SHOULD NOT REFER TO THE ENDOSCOPIC                    TREATMENT AS IDEAL. In case of UES "ideal" is the treatment that provides solution to the problem at the first attempt. In this case I would rather say that "Initial management of UES via endoscopic or percutaneous techniques may be considered/attempted".

Answer: As the reviewer suggested, we have modified the sentence (Page 6, Line 190–191).

              line 86: Robotic repair has also been considered as an option for the                              management of UES, but evidence is scarce on this topic

              line 90: percutaneous (not subcutaneous)

              line 94: the authors noted that 75% of patients were not cured after                              endoscopic or percutaneous approaches (more striking)

Page 7, line 99: ...remains unclear, robotic repair is an attractive option...

Answer: Thank the reviewer for the helpful suggestion. We have modified the sentence.

              line 103: rephrase, unclear (they showed that a total of 50 renal                                        units  were revised....WHAT DOES IT MEAN?).

Answer: We apologize for the unclear description. We have modified the sentence with clear description (Page 7, Line 218–220).

              LINE 117: Rephrase the Conclusions in a more effective way. For example: THIS STUDY HIGHLIGHTS THE CRUCIAL NEED TO PAY SPECIAL ATTENTION TO PREVENTION, EARLY DIAGNOSIS, METICULOUS REPAIR AND PROPER FOLLOW UP OF PATIENTS WITH UES AFTER  RARC, PARTICULARLY IN THE FIRST MONTHS AFTER SURGERY. IN CASE OF ILEAL CONDUIT, PARTICULAR ATTENTION SHOULD BE GIVEN TO THE LEFT URETERAL SIDE. IN CASE OF ORTHOTOPIC NEOBLADDER (FILL WITH YOUR OWN COMMENTS....). CAREFUL IDENTIFICATION AND IMPLEMENTATION OF STANDARDISED PROCEDURES, TECHNIQUES, TIPS AND TRICKS, AND METICULOUS REVIEW OF PERSONAL DATA AND EXPERIENCE ARE CRUCIAL IN DETECTING KEY REPRODUCIBLE POINTS TO INTRODUCE IN TRAINING AND SURGICAL PRACTICE, WITH THE AIM TO REDUCE THE INCIDENCE OF UES, A VERY SEVERE CLINICAL COMPLICATION AFTER RARC.

Answer: We would like to thank the reviewer for giving us clear and effective phrases for the Conclusion section. We modified the conclusion according to the reviewer’s comment and suggestion from an English editing service. (Page 7, Line 234–239).

Reviewer 3

Important issue
Ureteral stricture is under reported devastating complication and it is important that this review is clearly enlightening this issue.

Answer: We would like to thank the reviewer for the comments supporting our manuscript.

We would like to thank the reviewers for all the helpful comments and hope that we have now produced a more balanced and better account of our work. We hope that the revised manuscript is now suitable for publication in the Current Oncology. I herein confirm that all authors have agreed to the content of the revised version of this manuscript. Corresponding should be addressed to Shintaro Narita M.D. at the following address. Thank you in advance for your review of our work.

Shintaro Narita, M.D.

Department of Urology

Akita University School of Medicine

Hondo, Akita 010-8543, Japan

Fax: + 81-18-836-2619, Tel: + 81-18-884-6156

Reviewer 2 Report

This paper is interesting and brings attention to a very challenging and complex condition, such as UES after robotic radical cystectomy.  I like the fact that the title reports the 4 topics around which the paper is organised (Incidence, Etiology, Prevention and Management of UES), but I would change the wording: it is not DURING RARC, but AFTER RARC. Moreover the paper, presented as a review in the title, is not only a review, it reports also the original experience of the authors. So, as a first thing, I would change the title, and make it more correspondent to the text: INCIDENCE, ETIOLOGY, PREVENTION AND MANAGEMENT OF UES AFTER RARC: REVIEW OF PUBLISHED EVIDENCE AND PERSONAL EXPERIENCE.

A number of changes to the text are needed before the paper can be considered for publication, as shown below:

Abstract, line 9: Robot-assisted radical cystectomy (not prostatectomy)

                 line 10: ...can reach 25.3% of cases

Page 1, line 38: definitive conclusions

              line 41: this review focuses...

              line 42: ...after RARC, with the final aim to draw attention to this                                      severe clinical complication and to propose standardised                                    surgical steps able to reduce its incidence.

              line 44: may be possible/feasible (not ideal)

              line 45: ...have been observed only with surgical repair

Page 2, line 48: ...with an incidence rate ranging from 3 to 10%.

              line 56: ...given a very relevant clinical fact: 75% of patients with                                  UES are asymptomatic

Page 4, line 75: Particular care should therefore be taken with UES                                               prevention and diagnosis when dealing with the left ureter                                 and in the first months after RARC.

Page 4, line 84: but ICUD may be associated with less stretching of the                                        ureters and more accurate surgical technique thanks to                                      magnification

Page 4, line 86: ...the incidence rates of UES remain unclear considering the                             lack of...

Page 5, line 33 and following: why there is no mention on proposed standardised techniques in case of orthopic neobladders? Do the authors have experience on this? How should the anastomosis be done? How long the spatulation of the ureters in this case? Which technique? Nothing to add on this point? Or is the paper mainly focused on ileal conduits? If so, this should be specified in the title. Otherwise, the authors should mention trips and tricks in case of orthotopic neobladders, published evidence and personal experience.

Page 5, line 47: RARC (not RALC) 

              line 49: and preservation of periureteral tissues...

Page 6, line 52: a Total of 24 and 12 patients with ICUD urinary diversion                                     (PLEASE SPECIFY WHICH ONE: ILEAL CONDUIT?                                                NEOBLADDER?). THIS IS A VERY CRUCIAL POINT.

              line 73:...due to its higher success rate as compared...

              line 76: YOU SHOULD NOT REFER TO THE ENDOSCOPIC                    TREATMENT AS IDEAL. In case of UES "ideal" is the treatment that provides solution to the problem at the first attempt. In this case I would rather say that "Initial management of UES via endoscopic or percutaneous techniques may be considered/attempted".

              line 86: Robotic repair has also been considered as an option for the                              management of UES, but evidence is scarce on this topic

              line 90: percutaneous (not subcutaneous)

              line 94: the authors noted that 75% of patients were not cured after                              endoscopic or percutaneous approaches (more striking) 

Page 7, line 99: ...remains unclear, robotic repair is an attractive option...

              line 103: rephrase, unclear (they showed that a total of 50 renal                                        units  were revised....WHAT DOES IT MEAN?).

              LINE 117: Rephrase the Conclusions in a more effective way. For example: THIS STUDY HIGHLIGHTS THE CRUCIAL NEED TO PAY SPECIAL ATTENTION TO PREVENTION, EARLY DIAGNOSIS, METICULOUS REPAIR AND PROPER FOLLOW UP OF PATIENTS WITH UES AFTER  RARC, PARTICULARLY IN THE FIRST MONTHS AFTER SURGERY. IN CASE OF ILEAL CONDUIT, PARTICULAR ATTENTION SHOULD BE GIVEN TO THE LEFT URETERAL SIDE. IN CASE OF ORTHOTOPIC NEOBLADDER (FILL WITH YOUR OWN COMMENTS....). CAREFUL IDENTIFICATION AND IMPLEMENTATION OF STANDARDISED PROCEDURES, TECHNIQUES, TIPS AND TRICKS, AND METICULOUS REVIEW OF PERSONAL DATA AND EXPERIENCE ARE CRUCIAL IN DETECTING KEY REPRODUCIBLE POINTS TO INTRODUCE IN TRAINING AND SURGICAL PRACTICE, WITH THE AIM TO REDUCE THE INCIDENCE OF UES, A VERY SEVERE CLINICAL COMPLICATION AFTER RARC.

Author Response

(The authors gave the same response as above.)

Reviewer 3 Report

Important issue
Ureteral stricture is under reported devastating complication and it is important that this review is clearly enlightening this issue.

Author Response

(The authors gave the same response as above.)

Round 2

Reviewer 2 Report

I think the paper has been considerably improved after the first revision. It is for me now suitable for publication in its present form